# Using big data to search for possible geographic clustering of Congenital Heart Disease (CHD) across Australia

Calum Nicholson[1,2,3]*, Geoff Strange[1,2,3], David S. Celermajer[1,2,3]

**1** University of Sydney, Camperdown, New South Wales, Australia, **2** Heart Research Institute, 7 Eliza St, Newtown New South Wales, Australia, **3** Royal Prince Alfred Hospital, 50 Missenden Rd, Camperdown, New South Wales, Australia

* calum.nicholson@hri.org.au

## Abstract

Several diseases show geographic clustering, giving insights into possible genetic and environmental causes. The pathogenesis of Congenital Heart Disease (CHD) remains largely unknown and analysis of geographic distribution of CHD cases lacks input from large, national-scale datasets. People with structural CHD were selected from the Australia and New Zealand CHD Registry. Of people known to be still living, from linkage with the National Death Index, addresses were geocoded and aggregated to standardised geographic regions with measures of the Australian population. Areas were described based on measures of their remoteness and driving time to hospitals. The relationship between the distribution of the CHD and Australian populations was compared with bivariate spatial correlation. Of 81,349 people with structural CHD in the Registry, 63,863 were still living and could be geocoded. Overall, most people lived in Major Cities, and within 1-hour drive from a hospital, with the proportion the same across the CHD population, the "complex CHD" population and the Australian population. Across the country, there was a strong positive correlation between the Australian population and the CHD population. There were only a small number of areas (6%) where the Australian and the CHD populations were proportionally different. Overall, there was clear evidence that the geographic distribution of the CHD population proportionally follows the general Australian population. This suggests that there is unlikely to be any spatial clusters that are driven by genetic or environmental causes.

### Author summary

Outcomes for people living with congenital heart disease have improved greatly over recent decades. As surgical intervention has improved, people with congenital heart disease are living longer and a greater proportion are now adults. This

**Data availability statement:** Data access for the private health information used in this study is governed by Australian privacy legislation, which prevents the data being made publicly available. Furthermore, the geographic nature of this data makes true deidentification not possible. Access to this data requires approval from our governing ethics body, Sydney Local Health District Ethics Review Committee (RPAH Zone) (EC00113). Please contact the corresponding author, or the Registry team at admin@chaanz.org.au, for assistance in obtaining data access. Data from the Australian Census data can be obtained from the Australian Bureau of Statistic's Table Builder platform, however an institutional account or a paid personal account is required. Australian spatial data from the Australian Statistical Geographic Standard is publicly available, more information is available here: https://www.abs.gov.au/statistics/statistical-geography/australian-statistical-geography-standard-asgs.

**Funding:** This work was funded by the Australian Federal Government through the Medical Research Futures Fund The program was an "Accelerated Research Grant – congenital heart disease stream" (ARGCHDG000028 to DC and GS). This content is solely the responsibility of the authors and does not represent the official views of the funder. The funder had no role in study design, data collection and analysis, decision to publish, or preparation of the manuscript.

**Competing interests:** The authors have declared that no competing interests exist.

is success brings new challenges surrounding their healthcare. What kinds of complications will older people with congenital heart disease face, how will our health services cope with the increasing demands, and how should we deploy health services? We aimed to answer some of these question by assessing where people with congenital heart disease lived in Australia, and how that distribution compares with the general Australian population. This research is made possible by the Australia and New Zealand Congenital Heart Disease Registry, which enables this analysis to be conducted at a national scale for the first time. Most of the congenital heart disease population was living in major cities, and within a 1-hour drive of a hospital. Overall, their geographic distribution was very similar to that of the Australia population. These results suggest that there are not any environmental factors that are causing congenital heart disease, or that people with congenital heart disease are choosing to live in different places, compared to the general Australian population.

## 1. Introduction

With an estimated birth prevalence of 8 – 10 in 1,000 live births, Congenital Heart Disease (CHD) is the commonest birth defect [1,2]. There have been large demographic changes over the last half century, individuals with complex CHD are now surviving to 16 years at a rate of 87.5% [3]. Improved survival and an ageing CHD population has been well documented, with life expectancy of those with mild and moderate CHD matching the general population [4] resulting in the number of surviving adults with CHD greater than the number of children [5]. This ageing population creates new questions surrounding arising unknown complications. Chronic disease, comorbidities that increase with age, and the interactions between CHD and acquired heart disease are currently unknown challenges facing people with CHD [6,7]. The changing demographics of people with CHD exposes a gap in the literature surrounding current trends in the basic epidemiology of this population.

The aetiology of CHD is also not well understood. Whilst there have been advances in understanding the genetic risk factors associated with CHD, the causes of at least half of all CHD cases is still unknown [8]. Questions of how genetic risks might be linked to ethnic disparities are also being explored. Evidence suggests that the birth prevalence of certain types of CHD may differ between different racial groups [9,10], but any potential genetic or environmental mechanisms have not been described. To add to the genetic risk factors, environmental impacts are being explored as well. Links to maternal exposures such as pregestational diabetes [11], alcohol consumption, [12,13] and/or smoking [14] have all been investigated. Whilst gaps in knowledge remain, this evidence shows a complex, multifactorial aetiology of CHD.

Spatial epidemiology is a tool for understanding how genetic and environmental factors might impact the geographic distribution of a disease population. It is recognised that certain diseases have strong geographic clustering, with associations

to environmental factors identified in diseases such as lymphoma and amyotrophic lateral sclerosis [15,16]. We used the Australia and New Zealand CHD Registry (ANZCHD Registry - with now over 80 000 CHD cases) to understand the spatial distribution of people with CHD by identifying any clusters of CHD cases that might be out of proportion to the general Australian population. Identifying outlying clusters may focus future questions around the environmental and genetic factors influencing CHD in Australia.

We aimed, therefore, to study CHD case distribution across Australia, and compare that to the distribution of the total population, nationwide.

## 2. Materials and methods

### 2.1. Ethics statement

Human Research Ethics Approval was obtained for the ANZCHD Registry, from the Sydney Local Health District Ethics Review Committee (RPAH Zone) (EC00113). The number for this ethics protocol is 2019/ETH07472. Due to the large number of patients involved and the low-risk nature of the research, a waiver of consent was granted to collect retrospective patient data. Prospective data collection is approved via an opt-out consent process.

### 2.2. Participant selection

Participants were collected from the ANZCHD Registry, the data collection methods for this dataset have been described previously [17]. Briefly, data were collected from clinical databases and electronic medical records from participating hospitals in Australia and New Zealand. CHD cases were selected by presence of CHD diagnosis or procedure codes. Records that did not meet the minimum dataset or opted out of participation were excluded. We included those present in the Registry, who were living and had an Australian address that could be linked to a geocoded point. Only patients with structural CHD were included in the analysis, those with only genetic cardiac issues, or normal variants such as an isolated patent foramen ovale were excluded.

### 2.3. Data preparation

To convert participants' addresses to a longitude and latitude point, geocoding was completed using the ArcGIS geocoding service. Geocoded points were then standardised onto the Australia Statistical Geographic Standard (ASGS) by intersecting each point with a Statistical Area 1 (SA1) geographic area [18]. SA1 areas can be cleanly aggregated into larger "Statistical Areas" provided by the ASGS. For most of this analysis, Statistical Area 2 (SA2) is used to match with other publicly available datasets described below. CHD complexity was determined using a bespoke automated algorithm of diagnosis and procedure codes, following the European Society of Cardiology Guidelines for Adult Congenital Heart Disease [19,20].

The Accessibility and Remoteness Index of Australia (ARIA, 2021) was used to measure the accessibility and remoteness of geographic regions (each SA2) in Australia [21]. This remoteness measure calculates the percentage of the population in each SA2 area that live across five categories, "Major Cities", "Inner Regional", "Outer Regional", "Remote" and "Very Remote". The category with the largest association in an area was assigned to that area. The total Australian population for each SA2 area was taken from the 2021 Australian census and downloaded from the Australian Bureau of Statistics' Table Builder [22].

Hospital Travel Times dataset provides the driving time between each hospital (including description) in Australia and each SA2 area in Australia [23]. To measure the proximity of each SA2 area in Australia to its nearby hospitals, hospitals were described in three categories. These were hospital with an emergency room, hospitals with more than 200 beds, and hospitals described with the words "Large", "Major", or "Medium". Three true/false outcomes were developed, measuring whether any hospital from the corresponding category with within a 1-hour drive of that area.

PLOS Digital Health

Correlation between the geographic distribution of the general Australian population and the CHD population in the ANZCHD Registry was statistically determined (described below). These correlations were described through five categories. "High Australian Population and High CHD Population", and "Low Australian Population and Low CHD Population", where the CHD population was normally associated with the Australian population and there is no difference in proportion of population. "High Australian Population and Low CHD Population", where the relationship is negatively associated and the CHD population is proportionally lower than the Australian population. "Low Australian Population and High CHD Population", where the relationship is positively associated and the CHD population is proportionally higher than the Australian population. Finally, "Not Significant", where there is no matched association and there is no statistical evidence for a relationship between the CHD population and the Australian population.

### 2.4. Data analysis and software

Bivariate Spatial Correlation, Lee's L Statistic, was used to describe the relationship between the geographic distribution of the Australian Population and the CHD Population [24]. This technique measures the correlation between two observations, like a Pearson's Coefficient, but also considered the spatial association by calculating weighted values of each geographic feature's "neighbourhood". A neighbourhood of each geographic feature (the SA2 area) was determined by identifying all other SA2 areas with a contiguous boundary. "Equal Weighting" was used to create an average weight for each neighbourhood member by summing the total population across the neighbourhood and then dividing by the number of areas in the neighbourhood.

The analysis provides a global coefficient that outlines the overall bivariate association across all geographic features. This was used to answer our primary aim, to study CHD case distribution across Australia, and compare that to the distribution of the total population, nationwide. A Monte Carlo simulation (n = 10,000) was used to generate a pseudo-P-value to determine statistical significance, to a confidence of 95% [25]. Two other measures provide evidence for bivariate correlation. A spatial smoothing scalar measure autocorrelation in each observed variable. Values close to 1 suggest positive autocorrelation and values close to 0 suggest negative positive correlation. The second additional metric is the Pearson's correlation of the neighbourhood weighted averages of the two analysis variables. For example, a significant p-value where the spatial smoothing scalar is close to 1, the Pearson's correlation of the neighbourhood weighted averages is close to 0 and the global coefficient is close to 0, might suggests a result is due to autocorrelation rather than bivariate correlation.

In addition to the primary finding, we sought to explore the local variations that exists within the main findings of the global correlation by describing the correlation in each SA2 area. A local coefficient that provides a bivariate association for each geographic region was also calculated and a Monte Carlo Simulation was also performed to determine statistical significance, to a confidence of 95%.

All analysis were completed using R Statistical Software (v 4.4.2, 31/10/2024) [26]. Data cleaning was completed using the *tidyverse* package (v 2.0.0) [27], Spatial Analysis was completed with the *simple features* (*sf,* v 1.0.19) and *Spatial Dependence: Weighting Schemes, Statistics* (*spdep,* v1.3.9) packages [28,29]. Shapefiles used for mapping and spatial analysis are from the ASGS, accessed from the Australia Bureau of Statistics (ABS). ABS data is available for use under a Creative Commons Attribution (4.0 International), allowing free use of this data with appropriate attribution.

## 3. Results

### 3.1. Participant selection and cohort overview

There were 81,349 individuals with a structural CHD diagnosis selected from the Registry. Of the 76,655 who were still alive, 63,863 were successfully geocoded to an address in Australia, and matched to an SA2 area. 48% were male, 46% were female, less than 1% neither male or female, and there were 6% whose sex was missing or unknown. 21% of people

were born before 1990, 22% were born between 1990 – 1999, 24% were born between 2000 – 2009, and 33% were born between 2010 – 2024.

There were 32%; 38%; and 17% with mild, moderate and severe CHD respectively, with 13% whose severity was unable to be determined with automated processes. The most common CHD diagnoses were Ventricular Septal Defect (25%), Atrial Septal Defect (10%), Aortic Valve Disorders (9%), Coarctation of the Aorta (8%), and Persistent Arterial Duct (7%), Tetralogy of Fallot (6%), Pulmonary Valve Disorders (6%), and Transposition of the Great Arteries (5%). The full demographic characteristics are described in Table 1, stratified by remoteness area.

### 3.2. Correlation between Australian Population and CHD Population

Across the whole country, there was a positive spatial correlation between distribution of the total Australian population and the distribution of the CHD population (Lee's L Statistic: 0.40, p < 0.001). The spatial smoothing scalars were 0.51 and 0.40 for the Australian population and the CHD population respectively, suggesting no autocorrelation. The Pearson's correlation of the neighbourhood weighted averages was 0.78, also suggesting a positive bivariate correlation between the distribution of the total Australian population and the distribution of the CHD population.

When assessing the local spatial correlation for each (Fig 1), 76% of SA2 areas had no matched association between the total Australian and the CHD population (n = 1,872, p > 0.05). 18% of SA2 areas had a normal association (n = 427). Of these, 8% had a high Australian population and a high CHD population (n = 194), and 9% where both populations were low (n = 233). Only a small percentage of SA2 areas where negatively or positively associated (6%, n = 151), with 3% with a high Australian population and a low CHD population (n = 63) and 3% with a low Australian population and a high CHD population (n = 82).

### 3.3. Description of geographic distribution

The number of people who were living in the different ARIA remoteness areas, and within a 1-hour drive from various hospital types was counted. People with moderate or severe CHD were assessed separately to determine specific effects of more complex CHD.

The distribution of people living in various remoteness areas was largely similar for all three groups. Most people living within a "Major Cities of Australia" SA2 area, with 72%, 73%, and 72% from the total Australian population (n = 18,375,748), the CHD population (n = 46,283), and the complex CHD population (n = 25,681) respectively. The proportion of people from all three groups was also very similar for all other remoteness areas, with 17% to 18% in "Inner Regional Australia", 8% in "Outer Regional Australia", 2% in "Remote Australia" and 1% in "Very Remote Australia" (Fig 2a).

The proportion of people living outside of a 1-hour drive from various hospital types was very similar and small. 4% of people were living longer than a 1-hour drive from hospitals with an emergency department from all three groups (Fig 2b). Between 13% and 15% from all three groups were longer than 1 hour drive from hospitals with 200 beds or more (Fig 2c). Between 7% and 8% from all three groups were longer than 1 hour drive from hospitals described as "major", "large", or "medium" (Fig 2d).

### 3.4. Describing the positively and negatively associated areas

There was a very small proportion of SA2 areas that were positively or negatively associated. The local correlation categories of these rare outliers also compared against the remoteness areas and by driving times to the different hospital types (Fig 3). In SA2 areas with a high Australian population and a low CHD population, most areas were in Major Cities (89%) and within 1 hour drive of hospitals with an emergency room (94%), 200 beds or more (94%), and hospitals described as "major", "large", or "medium" (97%). There were some SA2 areas with a low Australian population and a

Table 1. Demographic overview of the congenital heart disease (CHD) cohort from the Australia and New Zealand CHD Registry, overall and stratified by remoteness area. Numbers less than ten have been masked as "<10" to maintain privacy, an (%).

| Characteristic | Overall N = 63,863[a] | Inner Regional Australia N = 10,827[a] | Major Cities of Australia N = 46,311[a] | Outer Regional Australia N = 4,943[a] | Remote Australia N = 1,065[a] | Very Remote Australia N = 717[a] |
|---|---|---|---|---|---|---|
| **Sex** | | | | | | |
| Female | 29,469 (46%) | 5,066 (47%) | 21,399 (46%) | 2,258 (46%) | 471 (44%) | 275 (38%) |
| Male | 30,475 (48%) | 5,441 (50%) | 21,912 (47%) | 2,388 (48%) | 453 (43%) | 281 (39%) |
| Other/Unknown | 3,919 (6%) | 320 (3%) | 3,000 (6%) | 297 (6%) | 141 (13%) | 161 (22%) |
| **Decade of Birth** | | | | | | |
| <1980 | 5,703 (9%) | 1,186 (11%) | 4,032 (9%) | 411 (8%) | 44 (4%) | 30 (4%) |
| 1980-1989 | 7,977 (13%) | 1,510 (14%) | 5,713 (12%) | 597 (12%) | 111 (10%) | 46 (6%) |
| 1990-1999 | 13,977 (22%) | 2,408 (22%) | 10,121 (22%) | 1,079 (22%) | 234 (22%) | 135 (19%) |
| 2000-2009 | 15,290 (24%) | 2,545 (24%) | 10,950 (24%) | 1,258 (25%) | 310 (29%) | 227 (32%) |
| 2010-2019 | 16,488 (26%) | 2,516 (23%) | 12,226 (26%) | 1,259 (25%) | 282 (27%) | 205 (29%) |
| 2020-2024 | 4,310 (7%) | 649 (6%) | 3,169 (7%) | 336 (7%) | 83 (8%) | 73 (10%) |
| **CHD Complexity** | | | | | | |
| mild | 20,052 (31%) | 3,298 (30%) | 14,517 (31%) | 1,600 (32%) | 380 (36%) | 257 (36%) |
| moderate | 24,510 (38%) | 4,125 (38%) | 17,850 (39%) | 1,903 (38%) | 387 (36%) | 245 (34%) |
| severe | 11,005 (17%) | 2,048 (19%) | 7,844 (17%) | 842 (17%) | 152 (14%) | 119 (17%) |
| unknown | 8,296 (13%) | 1,356 (13%) | 6,100 (13%) | 598 (12%) | 146 (14%) | 96 (13%) |
| **State** | | | | | | |
| ACT | 816 (1%) | 19 (0%) | 793 (2%) | <10 (<1%) | <10 (<1%) | <10 (<1%) |
| NSW | 21,978 (34%) | 3,891 (36%) | 16,912 (37%) | 1,010 (20%) | 126 (12%) | 39 (5%) |
| NT | 713 (1%) | <10 (<1%) | <10 (<1%) | 307 (6%) | 256 (24%) | 148 (21%) |
| QLD | 9,590 (15%) | 1,976 (18%) | 6,361 (14%) | 1,045 (21%) | 114 (11%) | 94 (13%) |
| SA | 6,700 (10%) | 552 (5%) | 5,076 (11%) | 811 (16%) | 194 (18%) | 67 (9%) |
| TAS | 1,153 (2%) | 646 (6%) | <10 (<1%) | 470 (10%) | 28 (3%) | <10 (<1%) |
| VIC | 14,767 (23%) | 3,176 (29%) | 10,867 (23%) | 685 (14%) | 20 (2%) | 19 (3%) |
| WA | 8,146 (13%) | 567 (5%) | 6,300 (14%) | 611 (12%) | 327 (31%) | 341 (48%) |

high CHD population found in regional or remote areas, or further away from hospitals. There were 41% of these SA2 areas across the Inner Regional, Outer Regional, Remote, and Very Remote remoteness areas. A small proportion were more than a 1-hour drive from hospitals with an emergency department (6%) and hospital described as "major", "large", or "medium" (11%), but a larger proportion were more than 1-hour drive from hospitals with 200 beds or more (31%).

## 4. Discussion

Overall, there is very little difference in the distribution of people with CHD compared to the general Australian population. Fig 2 shows no difference in the number of people with CHD living across the remoteness areas and within a 1-hour drive of hospitals compared to the Australian population. This was true even in those with moderate or complex CHD. A bivariate spatial correlation shows a strong positive correlation across the whole country (Lee's L Statistics: 0.4, p < 0.001). The spatial smoothing scalars suggest no autocorrelation that may be affecting the these results, providing further evidence for the strong positive correlation.

For the other 6% of SA2 areas, of particular interest are the "Low Australian Population and High CHD Population" areas where there are more people with CHD proportional to the Australian population. These areas occurred more often in more remote parts of the country. The number of these negatively associated areas is small, and they do not challenge the overall conclusion that the geographic distribution of CHD does not significantly differ from the

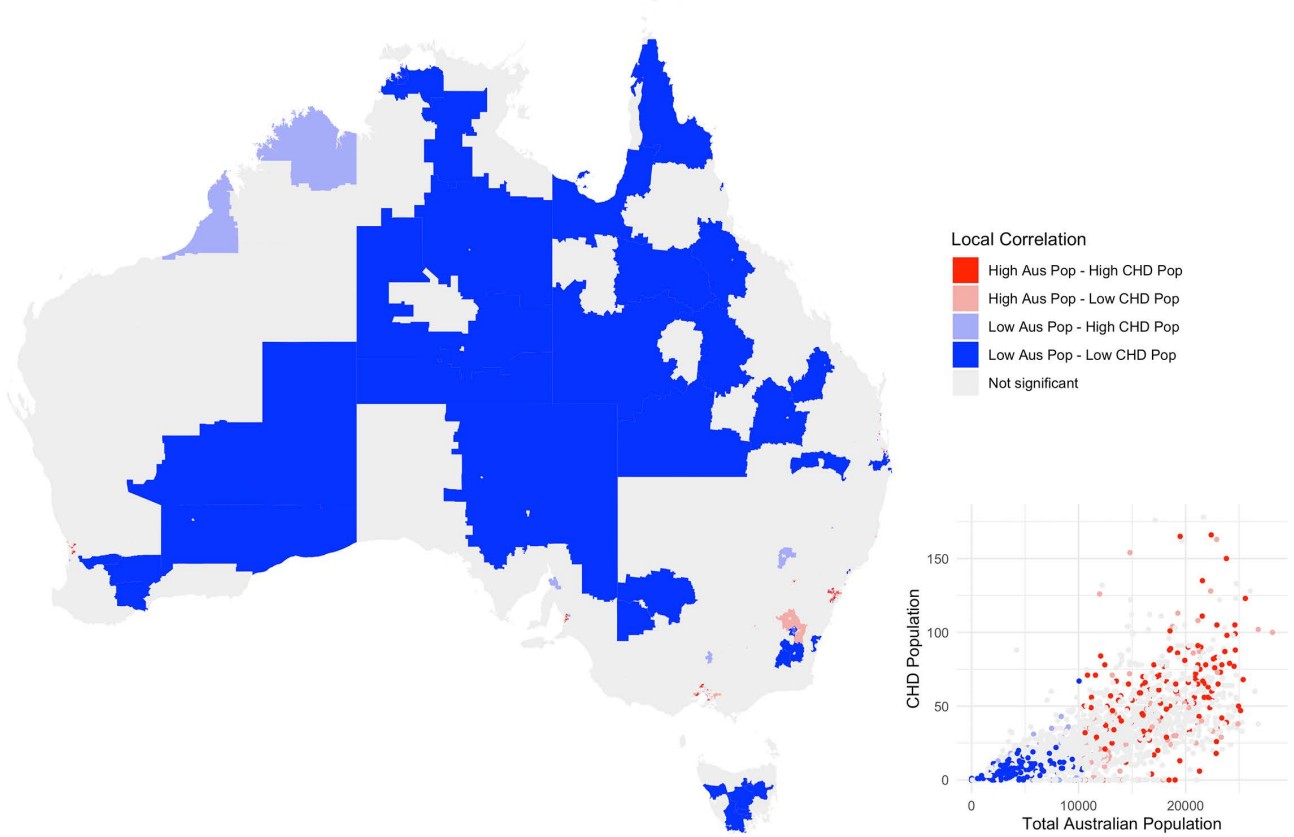

**Fig 1. Map of Australia showing the relationship between the general Australian population and the congenital heart disease (CHD) population.** The regions are "Statistical Area 2" (SA2), which is a standardized region designed by the Australian Bureau of Statistics with about 10,000 inhabitants. The relationship is described using five "local correlation" categories; "High Australian Population and High CHD Population", "High Australian Population and Low CHD Population", "Low Australian Population and Low CHD Population", and "Low Australian Population and High CHD Population", and "Not Significant". Inset shows a scatterplot of the corresponding Australian and CHD populations in each SA2 area, with the local correlation categories marked in colour. The base layer of this map uses SA2 shapefiles from the Australian Statistical Geography Standard, access from the Australia Bureau of Statistics (ABS). ABS data is available for use under a Creative Commons Attribution (4.0 International), allowing free use of this data with appropriate attribution.

Australian population. These local measures provide a useful description of the local variations that may occur within the overall global association, but care should be taken when assessing the significance of the repeated measures across each geographic reason. Further investigation is required to draw more conclusions about the variation at the local level.

These conclusions suggest that there are no clear genetic clusters, identifiable by SA2 areas and that there is no sign of toxicity exposures that are regionally specific. The spatial relationship between the CHD population and the Australia population shows no deliberate migration to urban areas, which we might expect to see if people with CHD are moving to areas with better access to specialised medical services. Australia has a highly urbanised population that is already close to medical services. Any small migration might not have been noticeable.

Disease clustering has been has been well established in many disease states, including cancers [30–32], Infectious disease [33], and amyotrophic lateral sclerosis [34–36]. Recently, Klein *et. al.* [37] have highlighted the importance of geospatial analysis in CHD, noting that current efforts are limited to small, single centre studies. We address this concern by providing a geospatial analysis on a national scale. Smaller studies have identified geographic clusters within their

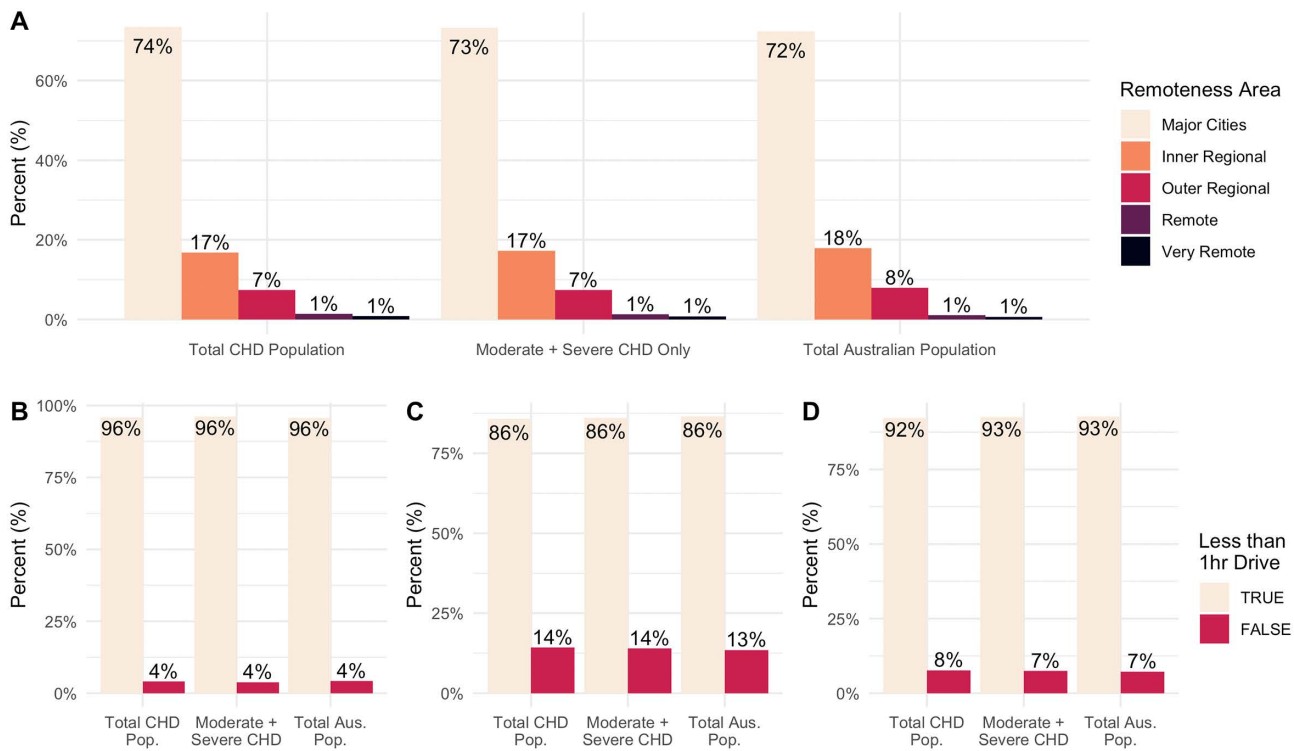

**Fig 2. Comparing the geographic distribution of the congenital heart disease (CHD) population, the complex CHD population (those with moderate and severe CHD), and the total Australian population.** Showing the proportion of patients living in each remoteness area **(A)** or living within a 1-hour drive of hospitals with emergency departments **(B)**, hospitals with 200 or more beds **(C)**, and hospitals described as "Major, "Large", or "Medium" **(D)**.

regions [38,39]. Though the causes could not be determined, their identification has been useful to guide resource allocation. Studies in China have noted potential geographic correlation in the Tuojiang River Basin with exposure risk to heavy metals [40] and in the Qinghai-Tibetan Plateau with CHD incidence correlated to higher altitudes [41]. Geographic difference has also been detected between pre- and postnatally diagnosed CHD, potentially highlighting inequitable access to CHD screening [42].

This analysis has been possible because of a large, national-scale, dataset of CHD patients. It is facilitated by linkage to publicly available data on the Australian population, including the Census and spatial data. Geocoding was an important here to match as many of the ANZCHD registry records as possible to standardised Australian geographies. The bivariate spatial correlation method is useful to compare between populations, providing more utility than univariate techniques. For example, methods such as spatial autocorrelation (Moran's I) or machine learning based clustering (e.g., k-means or DBscan), do not compare target populations against the geographic distribution of the whole population. With these techniques, it is not possible to ascertain whether identified clusters of disease populations are following the population density of the underlying population. Bivariate spatial correlation assesses whether a disease population's geographic distribution is out of proportion with the total population. The ability to assess both local as well as global spatial correlation, is also a strength of this analysis. With the local correlation values allows for some additional information to understand the variety that exists within the global trends.

There are some limitations to this study. A sensitivity analysis of the statistical methods to determine bivariate correlation would strengthen the results. First, different methods to determine neighbourhoods could be applied, include

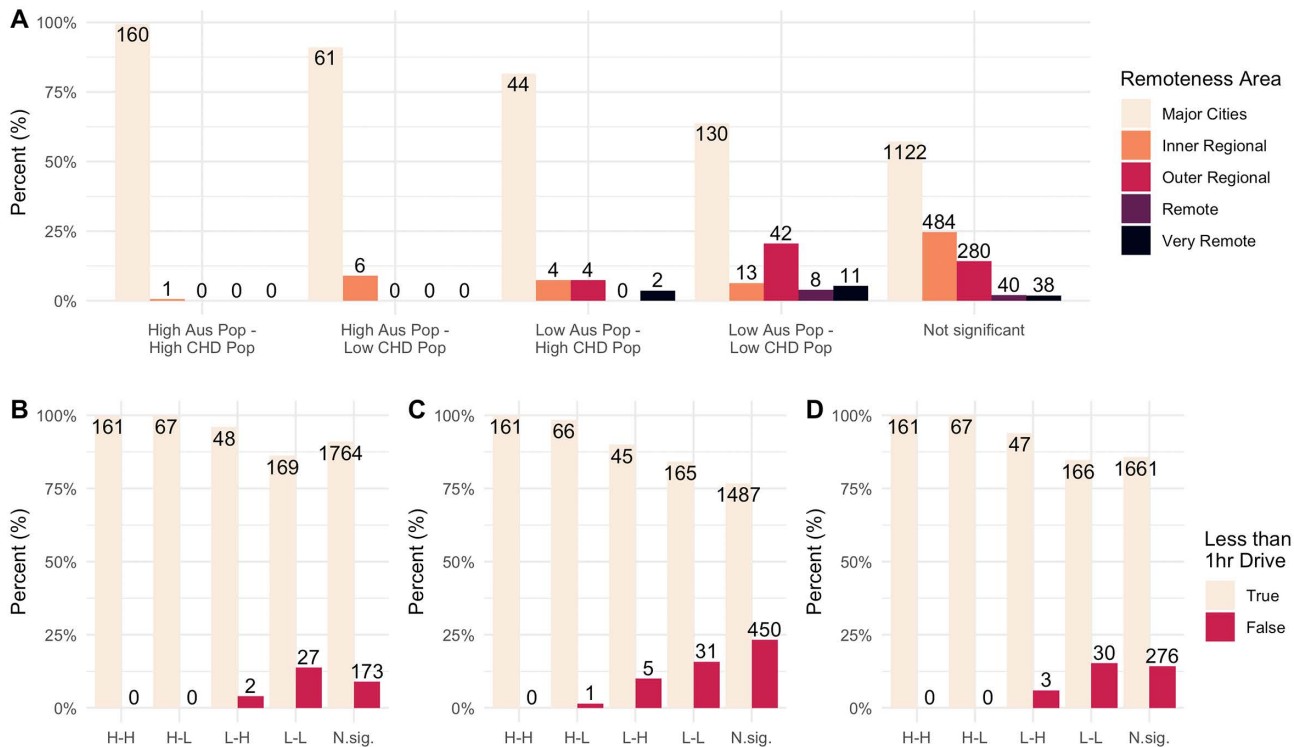

**Fig 3. Comparing how the local correlation categories are distributed amongst the remoteness areas (A) and the driving time hospitals with emergency departments (B), hospitals with 200 or more beds (C), and hospitals described as "Major, "Large", or "Medium" (D).** The local correlation categories are: "High Australian Population and High CHD Population" **(H-H)**, "High Australian Population and Low congenital heart disease (CHD) Population" **(H-L)**, "Low Australian Population and Low CHD Population" **(L-L)**, "Low Australian Population and High CHD Population" **(L-H)**, and "Not Significant" **(N.sig.)**. Numbers show the total number of Statistical Area 2 regions in each category. "Statistical Area 2" is a standardized region designed by the Australian Bureau of Statistics with about 10,000 inhabitants.

graph-based methods using the centre point of each SA2 area [43], or expanding the neighbourhood selection to include higher order neighbours (i.e., including neighbours of neighbours) [44]. There are also other statistical methods to assess bivariate spatial correlation, including Bivariate Moran's I Index [45], and Bivariate Spatial Cross-Correlation [46].

The comprehensiveness of data capture in the ANZCHD Registry should be considered. Whilst this is a large dataset, it is not a complete capture of the CHD population. Since data are collected from CHD services around Australia, it's patient population may be excluding patients who are lost to follow up or not currently known to specialist care. Patients who data are captured by the ANZCHD registry may be more likely to be closer to the CHD services who provide data. Given that rural outreach is routinely conducted in Australia, rural clusters where CHD populations are proportionally higher than the Australian population may be an artefact of rural outreach affecting patient capture in the Registry. Whilst it requires further exploration, this might suggest that the efforts to reach people with CHD living in rural areas has been successful in Australia.

Overall, this analysis shows a markedly positive spatial correlation between the CHD population and the general Australian population. These data suggest that "founder genetics" and clustered environmental causes are unlikely to be playing a major pathogenetic role in CHD, in our country.

## Author contributions

**Conceptualization:** Calum Nicholson, Geoff Strange, David S. Celermajer.

**Data curation:** Calum Nicholson.

**Formal analysis:** Calum Nicholson.

**Methodology:** Calum Nicholson, David S. Celermajer.

**Resources:** David S. Celermajer.

**Software:** Calum Nicholson.

**Supervision:** Geoff Strange, David S. Celermajer.

**Visualization:** Calum Nicholson.

**Writing – original draft:** Calum Nicholson, David S. Celermajer.

**Writing – review & editing:** Calum Nicholson, Geoff Strange, David S. Celermajer.

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
