## [Decision Letter · Decision Letter 0]

4 Nov 2025

Response to Reviewers
Revised Manuscript with Track Changes
Manuscript
**Journal Requirements:**

1. Please send a completed 'Competing Interests' statement, including any COIs declared by your co-authors. If you have no competing interests to declare, please state "The authors have declared that no competing interests exist".

3. Please provide separate figure files in .tif or .eps format.

4. Some material included in your submission may be copyrighted. According to PLOS’s copyright policy, authors who use figures or other material (e.g., graphics, clipart, maps) from another author or copyright holder must demonstrate or obtain permission to publish this material under the Creative Commons Attribution 4.0 International (CC BY 4.0) License used by PLOS journals. Please closely review the details of PLOS’s copyright requirements here: PLOS Licenses and Copyright. If you need to request permissions from a copyright holder, you may use PLOS's Copyright Content Permission form.

Potential Copyright Issues:

Figure 1: please (a) provide a direct link to the base layer of the map (i.e., the country or region border shape) and ensure this is also included in the figure legend; and (b) provide a link to the terms of use / license information for the base layer image or shapefile. We cannot publish proprietary or copyrighted maps (e.g. Google Maps, Mapquest) and the terms of use for your map base layer must be compatible with our CC-BY 4.0 license.

* U.S. Geological Survey (USGS) - All maps are in the public domain. (http://www.usgs.gov

**Additional Editor Comments:**

I would like to congratulate the authors for their work.

The manuscript is well writted and the methodology is robust.

However, some comments has to be adressed before potential publication.

?>**Reviewers' Comments:**

**Comments to the Author**

1. Does this manuscript meet PLOS Digital Health’s publication criteria?

Reviewer #1: Yes

Reviewer #2: Yes

2. Has the statistical analysis been performed appropriately and rigorously?

Reviewer #1: I don't know

Reviewer #2: Yes

3. Have the authors made all data underlying the findings in their manuscript fully available (please refer to the Data Availability Statement at the start of the manuscript PDF file)?

Reviewer #1: No

Reviewer #2: Yes

4. Is the manuscript presented in an intelligible fashion and written in standard English?

Reviewer #1: Yes

Reviewer #2: Yes

Reviewer #1: I commend the authors on their efforts, and thank them for the oportunity to review their work.

Summary:

This paper looks at patients with CHD in Australia, assessing if there are any geographic differences between the general population and this unique subgroup using data from national registries.

Quibbles:

There are quite a few stylistic issues I have for the abstract and authors summary, for example:

- First sentence of abstract has parentheses (cancer, neurological) after several disease.

These examples add nothing to the meaning and ruins the flow of the sentence.

- Line 31 "People living with congenital heart disease have overgone many changes over recent decades." People have overgone many changes? Or do you mean treatment options for people with CHD have changed over time?

I would suggest writing these clearer and with less editorialising.

The paper itself is easy to read.

The statistical tests used appear sound.

Major issue:

1. You tested for a statistical association across all SA2 areas. I undertand there are around 2,473, so setting a p-value of 0.05 means over a hundred or so will be statistically significance by chance alone without some kind of p-value correction. It's unclear if a bonferroni (or other) correction was applied. If not applied, could you provide a rationale as to why it wasn't.

Reviewer #2: The article analyzes the geographical distribution of congenital heart diseases (CHD) in Australia to eliminate potential geographical clusters that may reveal genetic or environmental causes. The statistical analysis of the ANZCHD registry through a bivariate spatial correlation (Lee's L statistic) of 63,863 individuals with cHD does not find any difference in the distribution of CHD within the general population.

The research question is legitimate and the possible implications for public health are significant.

The selected sample is large, but the registry used may introduce a selection bias excluding patients not followed in specialized services, or patients lost to follow-up.

The statistical analysis used allows for the definition of global and local spatial correlation, which answers the asked question, but other clustering methods using machine learning (k-means or DBscan) could have provided additional information independent of population density.

**Do you want your identity to be public for this peer review?** For information about this choice, including consent withdrawal, please see our Privacy Policy

Reviewer #1: No

Reviewer #2: No

**Figure resubmission:**

**Reproducibility:**To enhance the reproducibility of your results, we recommend that authors of applicable studies deposit laboratory protocols in protocols.io, where a protocol can be assigned its own identifier (DOI) such that it can be cited independently in the future. Additionally, PLOS ONE offers an option to publish peer-reviewed clinical study protocols. Read more information on sharing protocols at https://plos.org/protocols?utm_medium=editorial-email&utm_source=authorletters&utm_campaign=protocols

---

## [Editor Report · Decision Letter 1]

4 Feb 2026

Using Big Data to search for possible Geographic Clustering of Congenital Heart Disease (CHD) across Australia

PDIG-D-25-00333R1

Dear Mr Nicholson,

We are pleased to inform you that your manuscript 'Using Big Data to search for possible Geographic Clustering of Congenital Heart Disease (CHD) across Australia' has been provisionally accepted for publication in PLOS Digital Health.

Best regards,

Jeremy Florence

Guest Editor

PLOS Digital Health

**Additional Editor Comments (if provided):**

I have no further comment.